# Energy Expenditure, Protein Oxidation and Body Composition in a Cohort of Very Low Birth Weight Infants

**DOI:** 10.3390/nu13113962

**Published:** 2021-11-06

**Authors:** Michela Perrone, Camilla Menis, Pasqua Piemontese, Chiara Tabasso, Domenica Mallardi, Anna Orsi, Orsola Amato, Nadia Liotto, Paola Roggero, Fabio Mosca

**Affiliations:** 1Neonatal Intensive Care Unit, Fondazione IRCCS Ca’ Granda Ospedale Maggiore Policlinico, 20122 Milan, Italy; camilla.menis@gmail.com (C.M.); pasqua.piemontese@policliico.mi.it (P.P.); chiaratabasso@gmail.com (C.T.); domenica.mallardi@mangiagalli.it (D.M.); anna.orsi@policlinico.mi.it (A.O.); orsola.amato@policlinico.mi.it (O.A.); nadia.liotto@policlinico.mi.it (N.L.); paola.roggero@unimi.it (P.R.); fabio.mosca@unimi.it (F.M.); 2Department of Clinical Sciences and Community Health, University of Milan, 20122 Milan, Italy

**Keywords:** preterm infants, body composition, resting energy expenditure, indirect calorimetry, substrate oxidation

## Abstract

The nutritional management of preterm infants is a critical point of care, especially because of the increased risk of developing extrauterine growth restriction (EUGR), which is associated with worsened health outcomes. Energy requirements in preterm infants are simply estimated, so the measurement of resting energy expenditure (REE) should be a key point in the nutritional evaluation of preterm infants. Although predictive formulae are available, it is well known that they are imprecise. The aim of our study was the evaluation of REE and protein oxidation (Ox) in very low birth weight infants (VLBWI) and the association with the mode of feeding and with body composition at term corrected age. Methods: Indirect calorimetry and body composition were performed at term corrected age in stable very low birth weight infants. Urinary nitrogen was measured in spot urine samples to calculate Ox. Infants were categorized as prevalent human milk (HMF) or prevalent formula diet (PFF). Results: Fifty VLBWI (HMF: 23, PFF: 27) were evaluated at 36.48 ± 0.85 post-conceptional weeks. No significant differences were found in basic characteristics or nutritional intake in the groups at birth and at the assessment. No differences were found in the REE of HMF vs. PFF (59.69 ± 9.8 kcal/kg/day vs. 59.27 ± 13.15 kcal/kg/day, respectively). We found statistical differences in the protein-Ox of HMF vs. PFF (1.7 ± 0.92 g/kg/day vs. 2.8 ± 1.65 g/kg/day, respectively, *p* < 0.01), and HMF infants had a higher fat-free mass (kg) than PFF infants (2.05 ± 0.26 kg vs. 1.82 ± 0.35 kg, respectively, *p* < 0.01), measured with air displacement plethysmography. Conclusion: REE is similar in infants with a prevalent human milk diet and in infants fed with formula. The HMF infants showed a lower oxidation rate of proteins for energy purposes and a better quality of growth. A greater amount of protein in HMF is probably used for anabolism and fat-free mass deposition. Further studies are needed to confirm our hypothesis.

## 1. Introduction

Preterm birth interrupts fetal development, generally during the most critical phase of growth.

Historically, optimal growth has been defined as growth that simulates the intrauterine growth of healthy fetuses of the same post-conceptional age [1]. In recent years, new evidence has shown how this definition is not entirely acceptable, especially regarding very low birth weight infants (VLBWI).

Villar and coworkers recommended that the growth pattern of preterm infants should no longer be compared with the intrauterine growth of healthy fetuses, but with the growth of infants of the same gestational age and sex using adequate growth charts [2].

This change of paradigm is due to the awareness that extrauterine life exposes infants to an environment which is significantly different from the intrauterine one, and this also has some influence on development and growth.

This is because, as healthy as they may be considered, preterm infants present a relative grade of immaturity of organs and physiological and metabolic functions [3].

Preterm infants are in a state of energy deficit at birth, which may continue even during the first weeks of life, and this depends mostly on the abrupt loss of placental nutritional support. Considering the concomitant increase in energy expenditure secondary to enhanced respiratory work, and the higher metabolic demand due to concomitant acute illness [4], ensuring proper growth through adequate nutrition is still a challenge for neonatal care providers [5,6,7,8].

Optimal nutrition has previously been defined as the “energy and protein intake needed to maintain lean body mass and bone density, maximize neurodevelopment and minimize complications” [9]. However, the high incidence of growth retardation in preterm infants worldwide is a demonstration of how the optimization of nutritional support is still difficult to achieve in this population [4,5,10].

Weight is the sum of fat-free mass and fat mass. While weight gain resulting from the increased deposition of fat-free mass is associated with brain growth and better neurodevelopment outcomes at 12 months of life [11,12], it is also known that the excessive deposition of fat mass increases the risk of developing metabolic disturbance and non-communicable diseases during childhood and adulthood [13].

Previous studies conducted at our center demonstrated how the use of fortified human milk in preterm infants is positively associated with a greater deposition of fat-free mass, compared to infants fed with formula milk [14,15]. This finding is probably related to the positive nitrogen balance in infants fed with human milk. Another important point of care is the evaluation of the precise energy and nutrient requirements of VLBWI, in order to define a personalized nutritional program, set for a single patient.

Energy expenditure is often derived from predictive formulae in clinical practice, but current evidence suggests that they are not precise in their evaluation of the energy needs of children and have not been validated in preterm infants [16,17]. Indirect calorimetry (IC) is a non-invasive method to measure energy expenditure in newborns [18,19]. Indeed, IC also provides information about the oxidation of substrates for energy production [20,21,22].

The poor use of this method in neonatal intensive care units (NICUs) is the result of the necessity to keep infants quiet during the exam, since the values of gases measured (oxygen consumed and carbon dioxide produced) are influenced by movements [23].

In a previous study, conducted at our center, we demonstrated that twenty-minute IC provides results which are as reliable as two- to six-hour IC, as described in the past. This makes the exam more applicable in clinical practice [19].

Knowledge about the real energy and nutrient metabolism of VLBWI allows the prescription of personalized nutrition to guarantee optimal growth for this population.

For this reason, we conducted a study which aimed to evaluate resting energy expenditure, substrate oxidation and body composition at term corrected age in VLBWI, in relation to different modes of feeding.

## 2. Materials and Methods

### 2.1. Study Design

A cross-sectional observational study was conducted. VLBWI were enrolled and evaluated between 36 and 39 post-conceptional weeks.

### 2.2. Study Population

Very low birth weight infants (birth weight ≤ 1500 gr) with a gestational age of ≤32 weeks, born between January 2019 and December 2020 and admitted to the Neonatal Intensive Care Unit at Fondazione IRCCS Ca’ Granda Ospedale Maggiore Policlinico in Milan, were enrolled between 36 and 39 post-conceptional weeks, after obtaining informed written consent from parents. The ethical committee of our institution approved the study protocol.

All subjects included were born with a weight appropriate for gestational age according to Intergrowth-21st growth charts [2] and in a stable clinical condition, without major comorbidities. No patients were supplemented with oxygen or were taking chronic medications that could interfere with energy expenditure or growth. Moreover, they had reached full enteral feeding, defined as a mean daily volume of 150 mL/kg, with a good nutrition tolerance. The subjects included in the study are reported in Figure 1.

### 2.3. Nutritional Practice

During hospitalization, all infants were fed according to internal procedures for nutrition in VLBWI [24].

Parenteral nutrition (PN) was started on the first day of life. The volume of PN was gradually increased during the first week of life, starting from 80–90 mL/kg on the first day, and reaching a volume of 150–180 mL/kg on the seventh day of life. PN bags were prescribed according to ESPGHAN recommendations for the nutritional requirements of preterm infants [25,26]. Specifically, when clinical and metabolic stability was reached, PN provided 60–65 kcal/kg/day with 3.5–4.5 g/kg/day of protein. When the enteral feed reached a quota of 50 mL/kg/day, PN infusion was progressively decreased until suspension.

Enteral feeding was started as soon as possible, preferably within 24 h of postnatal life. Fresh milk from the infant’s mother was the first choice for all infants; if milk from the infant’s mother was not available or insufficient, pasteurized donor human milk was used. When enteral intakes reached 80 mL/kg/day with a good gastrointestinal tolerance, a target fortification using multicomponent bovine-based fortifiers was started, in accordance with ESPGHAN recommendations for the nutritional requirements of preterm infants [25,26].

According to these recommendations [25,26], our goal was to reach an enteral energy intake between 110 and 160 kcal/kg/day, with a protein intake of 3.5–4.5 g/kg/day.

Nutritional analysis (macronutrient and energy content) of donor pasteurized human milk and fresh mother’s milk was performed using a mid-ray spectrometry human milk analyzer (Miris AB), while the nutritional content of the formula was derived from the nutritional table provided by the manufacturer.

A week before discharge, preterm formula with a caloric density between 0.75 and 0.82 kcal/mL was gradually administered instead of pasteurized donor human milk. This shift became necessary because donor human milk would not be available after discharge. Infants were considered as human milk fed (HMF) if ≥75% of the daily volume was represented by human milk, and as formula fed (PFF) if ≥75% of the daily volume was represented by preterm formula. Nutritional intakes were recorded from computerized clinical charts (Neocare), and energy and macronutrient intakes (kcal/kg/day and g/kg/day, respectively) were calculated.

### 2.4. Clinical Data Collection

Infants’ baseline characteristics and mode of feeding were collected from patients’ computerized clinical charts.

### 2.5. Anthropometric Measurements

Weight, length and head circumference were measured by trained paramedical staff from our institution according to a standard procedure [27]. Weight was measured using the Seca Baby Scale 376. Length was measured to the nearest 1 mm on a Harpenden Infantometer. Head circumference was measured to the nearest 1 mm using non-stretch measuring tape [28]. Weight, length and head circumference z-scores were calculated using the z-score calculator provided by the Intergrowth-21st project [29].

### 2.6. Indirect Calorimetry

Indirect calorimetry was performed as previously described by Perrone and coworkers [19].

IC was conducted in a thermally controlled room immediately after a meal, with infants maintained in the supine position to keep them as calm as possible. Vital parameters were monitored continuously during the assessment. Moreover, no alteration in body temperature was detected. Resting energy expenditure (REE) was calculated by the measurement of oxygen consumed (VO2) and carbon dioxide produced (VCO2) through indirect calorimetry (IC) equipment, Quark RMR (COSMED-Italy), under steady state conditions. Steady state refers to a period of a minimum of five consecutive minutes during which the variation coefficients of VO2 and VCO2 values are lower than 10% [30]. Indirect calorimetry is based on the principle of the “open air circuit” system, which permits the measurement of VO2 and VCO2 using a constant flow generator. Expired air was sampled from a canopy placed around the baby. An external flowmeter permitted the adjustment of the flow rate during the exam. The respiratory quotient (RQ) is the ratio between carbon dioxide produced and oxygen consumed and reflects the utilization of different macronutrients as energy sources. In the resting condition, the RQ varies between 0.7 and 1, where a value of 1 reflects a prevalent utilization of carbohydrates as the energy source, while a value of 0.7 is associated with prevalent use of lipids [31].

The flow sensor and gas analyzers were calibrated every day before REE evaluation according to internal procedures. Calibration was considered acceptable if the variance coefficient was less than 1%.

REE measurements were performed for a mean period of 37 min, with a mean steady state period of 25 min. VO2 and VCO2 (mL/kg/min) were recorded every 10 s, and values were normalized using a neonatal convertor, provided by COSMED-Italy [19]. Behavioral states were evaluated using the modified Freymond Behavioral State Scale [23]. IC was stopped at a score of 2 or higher, which identifies an infant moving vigorously.

### 2.7. Substrate Oxidation

Considering that urinary nitrogen excretion is representative of total nitrogen loss, protein oxidation was calculated from daily urinary nitrogen excretion. A urine sample was collected on the day of IC using the cotton wool ball collection method, as described by Fell and colleagues [32]. Total daily urine output was computed by weighing diapers. Once the urinary nitrogen concentration (uN, g/L) was obtained, we derived protein oxidation (g/kg/day) by applying the formula “urinary nitrogen (g/kg/day) ∗ 6.25/kg of body weight”, where 6.25 is the conversion factor for obtaining protein from nitrogen.

Carbohydrate and lipid oxidation were calculated through the application of Frayn’s formulae that include the measurement of VO2 and VCO2 during IC. The mentioned formulae are as follows [20]:Carbohydrate oxidation (g/kg/day) = [4.55 ∗ VCO2 (mL/kg)] − [3.21 ∗ VO2 (mL/kg)] − 2.87 ∗ uN
Lipid oxidation (g/kg/day) = [1.67 ∗ VCO2 (mL/kg)] − [1.67 ∗ VO2 (mL/kg)] − 1.92 ∗ uN

### 2.8. Body Composition

Body composition was assessed using an air displacement plethysmography system (PEA POD Infant Body Composition System; COSMED SRL). The precision and accuracy of this method in the evaluation of infants’ body composition were previously confirmed by Roggero et al. [33].

The assessment of fat mass (FM) and fat-free mass (FFM) was performed through the application of a densitometric model, which takes into account body mass and body volume. Body mass was measured on an integrated electronic scale, and body volume was assessed in the PEA-POD test chamber by applying gas laws. Then, the percentage of body fat, defined as body weight minus fat-free mass, was calculated using body density and constant FM density values (set at 0.9007 g/mL), while FFM density values were calculated as the sum of the contribution of the various components in the fat-free mass compartment [34]. Age- and sex-specific fat-free mass density values deduced from data by Fomon et al. were used [35].

### 2.9. Statistical Analysis

Continuous variables are presented as the mean and standard deviation, while categorical variables are expressed as absolute numbers or percentages. Despite the limited number of subjects in the two groups, we used the mean and standard deviation (SD) instead of the median because the distribution of our population was normal, since the values of the mean and median coincide.

Considering the available data regarding REE in VLBWI according to the mode of feeding [36], we decided to enroll a minimum of 20 VLBWI to have an α-error = 0.05 and a power of 95%. REE, FFM, FM and nutritional intakes were compared by the SPSS 20 statistical package, using the t-test for unpaired data. A *p*-value of 0.05 was considered significant.

## 3. Results

Fifty VLBWI (HMF:23, PFF:27) were evaluated at 36.48 ± 0.85 post-conceptional weeks.

No significant differences were found in basic characteristics at birth or at indirect calorimetry, or in terms of nutritional intakes, in the two groups (Table 1 and Table 2, respectively).

In Table 3, data regarding REE and substrate oxidation are reported.

All infants maintained a behavioral state score lower than 2 according to the Freymond Behavioral State Scale. No differences were found in the REE of HMF vs. PFF (59.69 ± 9.8 kcal/kg/day vs. 59.27 ± 13.15 kcal/kg/day, respectively), while a statistical difference was found if REE was expressed as an REE/FFM ratio (64.18 ± 1.46 kcal/FFM/day vs. 73.31 ± 1.87 kcal/FFM/day in HMF and PFF, respectively, *p* < 0.001). We found a statistical difference in the protein oxidation of HMF vs. PFF (1.7 ± 0.92 g/kg/day vs. 2.8 ± 1.65 g/kg/day, respectively, *p* < 0.01). A difference, although not statistically significant, was found in the carbohydrate oxidation rate (7.28 ± 1.36 g/kg/day vs. 7.01 ± 1.97 g/kg/day in HMF and PFF, respectively, *p* = 0.56). No differences were found in lipid oxidation (1.27 ± 0.37 g/kg/day vs. 1.25 g/kg/day in HMF and PFF, respectively). The respiratory quotient, which is an indirect index of substrate oxidation, was significantly different between the two groups (0.89 ± 0.04 in HMF and 0.85 ± 0.08 in PFF, *p* < 0.05).

Lastly, significant differences were found in the body composition of the HMF and PFF groups.

In particular, HMF infants presented a higher fat-free mass (kg) than PFF infants (2.05 ± 0.26 kg vs. 1.82 ± 0.35 kg, respectively, *p* < 0.01). These differences were also significant if fat-free mass was expressed as a percentage of body weight. Data regarding body composition are presented in Table 4.

## 4. Discussion

In this study, we explored whether the energy metabolism and body composition of stable VLBWI could be influenced by the mode of feeding.

We did not find any significant difference in the REE of HMF vs. PFF infants in terms of kilocalories per kilogram of body weight, but we found that infants with a prevalent human milk diet had lower protein oxidation. A greater amount of protein in HMF is probably used for anabolism and the deposition of fat-free mass. Our hypothesis is supported by the evidence that, in our population, the prevalent human milk diet was associated with a greater percentage of fat-free mass at term corrected age.

In a previous study, Morlacchi et al. had already shown a positive correlation between being fed with human milk and an increased deposition of fat-free mass in VLBWI. Our data corroborate these hypotheses, as we found a mean of 2 kg of FFM in HMF vs. 1.82 kg of FFM in PFF infants (*p* < 0.01), without differences in weight between the groups at the time of assessment [14].

These data have been confirmed by other authors, who investigated the influence of the mode of feeding on body composition in preterm infants.

Larkade et al. reported a positive correlation between how long VLBWI are fed with human milk (in terms of the number of days) and fat-free mass content at hospital discharge [37]. Similarly, a meta-analysis conducted by Huang et al. concluded that preterm infants fed with formula have a greater fat mass than preterm infants fed with human milk [38].

A study performed by Giannì et al. showed similar results in late preterm infants. They demonstrated that being fed with human milk is associated with a greater deposition of fat-free mass at term corrected age in a population of healthy late preterm infants [15].

A prevalent human milk diet probably promotes tissue growth and fat-free mass deposition if nutritional needs are met. This phenomenon could be explained by the facilitation of intestinal absorption and utilization of macronutrients performed by the protein of human milk, as demonstrated by Lönnerdal and coworkers [39].

This allows us to speculate on the fact that HMF infants have a greater amount of carbohydrates and lipids for energy purposes, as evidenced by the lower protein oxidation found in our population fed with a prevalent human milk diet.

A previous study conducted by Lubetzky and coworkers analyzed, for the first time, the differences in the REE of preterm infants fed with human milk (HM) or formula milk (PF). Infants fed with human milk had lower REE compared with formula-fed infants (52 ± 6 vs. 57 ± 10 kcal/kg day in HM and PF, respectively), but this difference was not statistically significant. When data were reported as the mean REE, corrected for actual calories consumed by the infants, a significant difference was found. Indeed, HM infants had an REE/Kcal ratio of 0.71 ± 1.1, while PF infants had an REE/kcal ratio of 0.85 ± 1.6, *p* = 0.004) [36].

Mendes Soares conducted a similar analysis centered on differences in REE in preterm infants fed with HM or PF. They concluded that preterm formula produces a better metabolic response compared with human milk, but this difference disappears when human milk is fortified to reach an adequate amount of energy and protein [40].

A similar analysis was published in 1984 by Putet and collaborators [41].

They evaluated energy expenditure using IC and the oxidation of proteins in VLBWI according to the mode of feeding. Preterm infants were divided into two groups: non-fortified human milk diet (HM) vs. preterm formula diet (PF). The study population was evaluated at 36 weeks corrected age. They found a difference in the nutritional intakes (energy 109 ± 11.1 kcal/kg/day and protein 2.36 ± 0.03 g/kg/day in HM infants vs. energy 129.8 kcal/kg/day and protein 3.13 ± 0.1 g/kg/day in PF, *p* < 0.005 and *p* < 0.01, respectively.) The oxidation rate of proteins was significantly different at term corrected age (0.51 ± 0.1 g/kg/day in HM vs. 0.76 ± 0.2 g/kg/day in PF, *p* < 0.05). A significant difference was also found in REE (51.5 ± 2.9 kcal/kg/day vs. 63.3 ± 4.5 kcal/kg/day in HM vs. PF, *p* < 0.005). Growth parameters were also different, with a mean body weight of 1914 ± 75 g in HM and 2159 ± 87 g in PF (*p* < 0.002). In this case, the lower REE and lower protein oxidation in HM could be the consequence of the energy and protein intake being insufficient to cover the nutritional needs of preterm infants [41].

In our study, the basic characteristics and the nutritional intakes of infants fed with HM or PF were equal, meaning no influences of inadequate nutritional intake could confound our results.

While previous studies investigated protein balance according to the feeding regimen in preterm infants [14,41,42,43], to our knowledge, this is the first work analyzing substrate oxidation using IC and body composition in VLBWI. Our data show that infants on a prevalent human milk diet use carbohydrates as their preferred source of energy, with a relative preservation of proteins. No differences were found in lipid oxidation between the groups. As a result of this, HMF infants have a greater amount of protein available for anabolism and fat-free mass deposition.

The present study has some limitations. First, we included only healthy and clinically stable VLBWI, meaning the influences of comorbidities or complications on REE, body composition and substrate oxidation were not assessed. Furthermore, the evaluation of REE, body composition and substrate oxidation was performed only at discharge, meaning we did not investigate the influences of human milk on these variables over time.

The main strength of this paper is the large number of clinically stable VLBWI enrolled, which permitted us to obtain reliable data on REE, body composition and substrate oxidation under para-physiological conditions, without interference from other variables except the mode of feeding. Moreover, these data should prompt neonatal nutritional teams to provide adequate ratios of non-protein energy/grams of protein, in order to stimulate growth and fat-free mass deposition.

Further studies are necessary to evaluate the persistence of these modulating effects during the first months of life.

## Figures and Tables

**Figure 1 nutrients-13-03962-f001:**
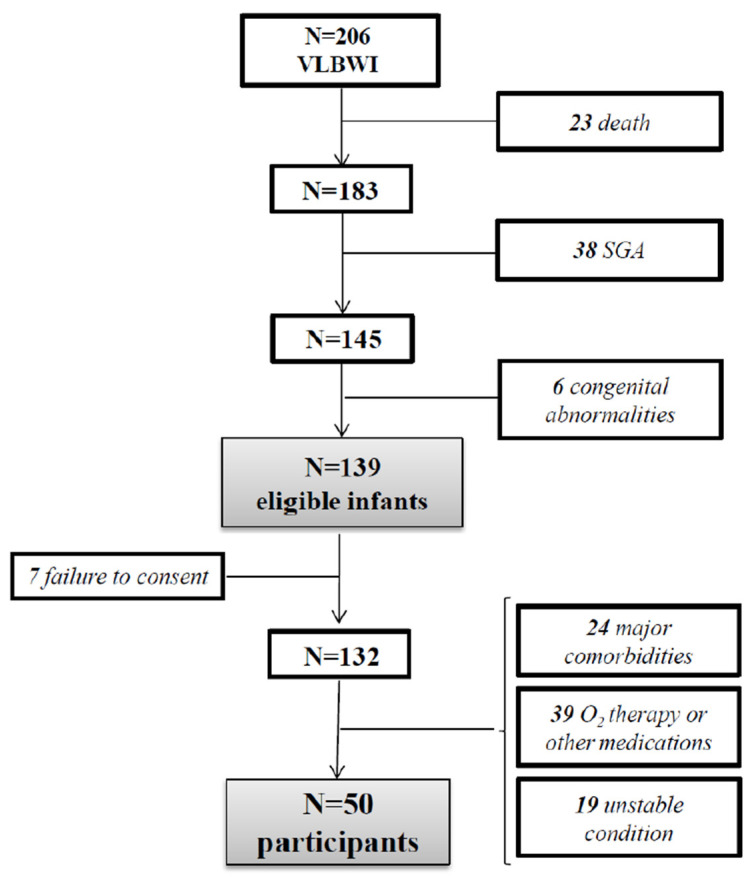
Flow chart explaining the recruitment process. VLBWI, very low birth weight infants.

**Table 1 nutrients-13-03962-t001:** Basic characteristics of subjects.

	HMF (23)			PFF (27)			
	Mean	SD	Range	Mean	SD	Range	*p*
GA (weeks)	30.09	2.02	(25–32)	30.33	1.44	(26–33)	ns
Birth weight (g)	1280.22	252.5	(720–1490)	1283.67	159.3	(890–1500)	ns
Birth weight z-score	−0.46	0.59	(−1.27–0.88)	−0.45	0.61	(−1.27–0.87)	ns
GA at IC (weeks)	36.48	0.85	(36–39)	36.7	0.99	(36–39)	ns
Weight at IC (g)	2156.22	331.3	(1800–3310)	2251.26	292.57	(1861–2795)	ns
Weight z-score at IC	−0.97	0.86	(−2.29–1.47)	−0.84	0.88	(−2.44–0.63)	ns

GA, gestational age; IC, indirect calorimetry.

**Table 2 nutrients-13-03962-t002:** Nutritional intakes at IC.

	HMF (23)			PFF (27)			
	Mean	SD	Range	Mean	SD	Range	*p*
Volume mL/kg/day	152.46	15.79	(116.95–184)	153.14	14	(125.99–182.92)	ns
Energy kcal/kg/day	126.69	19.33	(100.4–166.27)	125.55	12.37	(105.4–158.75)	ns
Protein g/kg/day	3.34	0.72	(2.14–4.66)	3.47	0.46	(2.59–4.39)	ns
Carbohydrate g/kg/day	13.18	1.91	(10.22–16.55)	13.3	1.74	(9.25–15.91)	ns
Lipid g/kg/day	7.08	1.14	(5.57–9.52)	7.56	0.73	(6.31–9.83)	ns

**Table 3 nutrients-13-03962-t003:** REE and substrate oxidation.

	HMF (23)			PFF (27)			
	Mean	SD	Range	Mean	SD	Range	*p*
REE kcal/kg/day	59.69	9.8	(47.01–82.71)	59.27	13.15	(45.42–105.09)	ns
REE kcal/FFM(kg)day	64.18	1.46	(62.28–65.65)	73.31	1.87	(71.40–75.21)	*p* < 0.001
RQ	0.89	0.04	(0.77–0.95)	0.85	0.08	(0.69–0.97)	*p* < 0.05
Protein ox g/kg/day	1.7	0.92	(0.41–3.66)	2.8	1.65	(0.9–6.34)	*p* < 0.01
Carbohydrate ox g/kg/day	7.28	1.36	(5.17–9.22)	7.01	1.97	(3.62–9.71)	*p* = 0.56
Lipid ox g/kg/day	1.27	0.37	(0.51–1.82)	1.25	0.49	(0.45–2.05)	ns

REE, resting energy expenditure; RQ, respiratory quotient; Ox, oxidation.

**Table 4 nutrients-13-03962-t004:** Body composition.

	HMF (23)			PFF (27)			
	Mean	SD	Range	Mean	SD	Range	*p*
FFM kg	2.05	0.26	(1.68–2.55)	1.82	0.35	(1.17–2.88)	*p* < 0.01
FFM %	88.02	3.93	(80.40–94.90)	84.91	4.12	(78.45–89.48)	*p* < 0.01
FM kg	0.28	0.12	(0.11–0.60)	0.34	0.09	(0.11–0.42)	*p* < 0.05
FM %	11.97	3.93	(5.1–19.6)	15.09	2.87	(7.12–21.4)	*p* < 0.01

FFM, fat-free mass; FM, fat mass.

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
