# Peer review of "Energy Expenditure, Protein Oxidation and Body Composition in a Cohort of Very Low Birth Weight Infants"

_nutrients, 2021, doi:10.3390/nu13113962_

Round 1
Reviewer 1 Report
This is an interesting observational study, the results add to the literature. The following points have to be addressed in a revised version of the ms.
First, the precision of the VO2- and VCO2-measurements in infants should be given. In addition, the calibration of the QuarkQMR should be described. To address this point the authors are referred to Obesity 2017;25:1941-1947 and Am J Clin Nutr 2013;97:763–73.
Second, there is need of a more systematic data analysis, i.e., the REE on FFM-association has to be addressed and the residuals have to be analyzed and compared between the two intervention groups. In addition, the variance in REE should be compared with the variance in body temperature to give an idea about the systemic outcome of metabolism.
Third, there is a fundamental flaw in data analysis and presentation, i.e., since the REE on body weight (or FFM) regression line has a non-zero intercept REE it is not allowed to divide REE by body weight (or FFM). To get that idea the authors are referred to Am J Clin Nutr. 1989; 49: 968–975.
Fourth, the authors have assessed pp-EE. Thus, the measurement includes REE and DIT. To address this point, the variance in the REE on FFM residuals should be compared with protein intake and protein balance, thus, the thermic effect of proteins can be addressed.
Author Response
This is an interesting observational study, the results add to the literature. The following points have to be addressed in a revised version of the ms.
First, the precision of the VO2- and VCO2-measurements in infants should be given. In addition, the calibration of the QuarkQMR should be described. To address this point the authors are referred to Obesity 2017;25:1941-1947 and Am J Clin Nutr 2013;97:763–73.
Thank you for your suggestion. We provided to add the calibration process. We have yet demonstrated the precision of VO2 and VCO2 in our precedent publication, so we reported that IC was performed as previously described by Perrone and coworkerd. (See Nutrition 2021;86:111180)
Second, there is need of a more systematic data analysis, i.e., the REE on FFM-association has to be addressed and the residuals have to be analyzed and compared between the two intervention groups. In addition, the variance in REE should be compared with the variance in body temperature to give an idea about the systemic outcome of metabolism.
Thank you for your suggestion. We provided to modify the text. We don’t have any variation on body temperature (we performed short-time IC), so we decided to add only a sentences which affirm that body temperature was normal. I hope that this sentences, in association with no variation in room temperature and vital signs is sufficient to exclude influence of temperature on REE.
Third, there is a fundamental flaw in data analysis and presentation, i.e., since the REE on body weight (or FFM) regression line has a non-zero intercept REE it is not allowed to divide REE by body weight (or FFM). To get that idea the authors are referred to Am J Clin Nutr. 1989; 49: 968–975.
Thank you for your comment. Actually, we didn’t perform a regression, but only t-test analysis as explain in statistical analysis section.
Fourth, the authors have assessed pp-EE. Thus, the measurement includes REE and DIT. To address this point, the variance in the REE on FFM residuals should be compared with protein intake and protein balance, thus, the thermic effect of proteins can be addressed.
Thank you for your comment. Actually we perform indirect calorimetry during the first 20 minutes after meal. As you known, DIT takes place when food is absorbed from the intestine and used for metabolic purposes. Considering that, no effect of DIT should be evident in such a short period of time, since in the first 20 minutes the metabolic process has not yet been started.

Reviewer 2 Report
Thank you for allowing me to review this manuscript. This is a timely and novel topic in the care of preterm infants. My comments and suggestions are as follows:
- Line 12: Consider changing “worse” to “worsened”.
- Line 13: Consider changing the connotation of this sentence as we do have suggested recommendations for intakes. Instead of stating “energy needs are not well known”, perhaps you can rephrase as “energy requirements in preterm infants are simply estimations, so the measurement of resting energy expenditure…”.
- Line 14-15: Please correct the grammar of this sentence. Please review the entire manuscript for appropriate English grammar.
- Line 16: Please change “substrates” to “protein”.
- Line 17: Please change “correlation” to “association”. Please note that your objective in the abstract appears different than the objective listed in your background/introduction.
- Line 39-41: Please consider rephrasing this statement. Perhaps the definition of appropriate growth in a preterm infant isn’t “invalid”, but more so that extrauterine variables must be considered in a preterm infant compared to a fetus growing in an intrauterine environment.
- Line 54: I agree that preterm infants can be at a nutrition deficit, but this does not always continue “in the early weeks of life”. May units provide aggressive enteral and/or parenteral nutrition, so adequate nutrition can be provided early in life (even in the first week of life).
- Line 59: Please change the term “neonatologists” to something like “neonatal care providers”. Many units provide comprehensive nutritional care by a team of multidisciplinary members.
- Line 62-65: Please consider that there are some unit who have published data demonstrating adequate growth with optimized nutrition practices, indicating it is possible to provide adequate nutrition.
- Line 77-79: This sentence indicates there are predictive equations for preterm infants, but the included references (16, 17) only included non-preterm infants and older aged children. Thus, please correct the accuracy of this sentence as I am unaware of equations for a preterm population.
- Line 102: Please define acronyms (e.g. “gr”) before using the abbreviation.
- Please provide more details of nutrition management, such as type of human milk fortifier or infan formula used, how parenteral nutrition is managed, what goal energy and protein intakes are, what caloric density was used, etc.
- Line 132-132: How would you classify infants who received half of feedings as human milk and half as preterm formula?
- I appreciate your use of appropriate methods of evaluation, such as air displacement plethysmography, etc.
- I appreciate your explanation for choosing to report mean vs. median.
- Please check with the journal’s (Nutrients) guidelines for use of decimal points vs. commas in published manuscript (e.g. Table 2, etc). For example, a standard deviation of “1,44” could be reported 1.44 in the table.
- In your discussion section, can you provide greater detail how clinicians can/should use values from the REE? This is resting energy expenditure (low at ~60 kcal/kg/day), but this is not adequate to promote appropriate growth. How would clinicians use to adjust and individual care for each infant? Likewise, how would clinicians use protein oxidation values in real-time clinical care?
- I agree with your reported limitations. A huge consideration (which you included) is that there was no inclusion of infants with chronic conditions (e.g. bronchopulmonary dysplasia, etc.) who would have altered needs.
Author Response
Line 12: Consider changing “worse” to “worsened”.
Thank you for your suggestion. We provided to modify the text.
Line 13: Consider changing the connotation of this sentence as we do have suggested recommendations for intakes. Instead of stating “energy needs are not well known”, perhaps you can rephrase as “energy requirements in preterm infants are simply estimations, so the measurement of resting energy expenditure…”.
Thank you for your suggestion. We provided to modify the text.
Line 14-15: Please correct the grammar of this sentence. Please review the entire manuscript for appropriate English grammar.
Thank you for your suggestion. We have programmed an English editing before final submission.
Line 16: Please change “substrates” to “protein”.
Thank you for your suggestion. We provided to modify the text.
Line 17: Please change “correlation” to “association”. Please note that your objective in the abstract appears different than the objective listed in your background/introduction.
Thank you for your suggestion. We provided to modify the text.
Line 39-41: Please consider rephrasing this statement. Perhaps the definition of appropriate growth in a preterm infant isn’t “invalid”, but more so that extrauterine variables must be considered in a preterm infant compared to a fetus growing in an intrauterine environment.
Thank you for your suggestion. We provided to modify the text.
Line 54: I agree that preterm infants can be at a nutrition deficit, but this does not always continue “in the early weeks of life”. May units provide aggressive enteral and/or parenteral nutrition, so adequate nutrition can be provided early in life (even in the first week of life).
Thank you for your suggestion. We provided to modify the text.
Line 59: Please change the term “neonatologists” to something like “neonatal care providers”. Many units provide comprehensive nutritional care by a team of multidisciplinary members.
Thank you for your suggestion. We provided to modify the text.
Line 62-65: Please consider that there are some unit who have published data demonstrating adequate growth with optimized nutrition practices, indicating it is possible to provide adequate nutrition.
Thank you for your suggestion. With this sentence we would say that it is difficult to guarantee an adequate energy and nutrients intake in this vulnerable population, so a special attention is requested.
Line 77-79: This sentence indicates there are predictive equations for preterm infants, but the included references (16, 17) only included non-preterm infants and older aged children. Thus, please correct the accuracy of this sentence as I am unaware of equations for a preterm population.
Thank you for your suggestion. We provided to modify the text.
Line 102: Please define acronyms (e.g. “gr”) before using the abbreviation.
Thank you for your suggestion.
Please provide more details of nutrition management, such as type of human milk fortifier or infan formula used, how parenteral nutrition is managed, what goal energy and protein intakes are, what caloric density was used, etc.
Thank you for your suggestion. We provided to modify the text.
Line 132-132: How would you classify infants who received half of feedings as human milk and half as preterm formula?
We divided infants according to mode of feeding: prevalent human milk diet versus prevalent formula diet, so we considered only infants with an amount of human milk or formula milk > 75%respectively.
I appreciate your use of appropriate methods of evaluation, such as air displacement plethysmography, etc.
Thank you for your comment.
I appreciate your explanation for choosing to report mean vs. median.
Thank you for your comment.
Please check with the journal’s (Nutrients) guidelines for use of decimal points vs. commas in published manuscript (e.g. Table 2, etc). For example, a standard deviation of “1,44” could be reported 1.44 in the table.
Thank you for your suggestion. We provided to modify the text.
In your discussion section, can you provide greater detail how clinicians can/should use values from the REE? This is resting energy expenditure (low at ~60 kcal/kg/day), but this is not adequate to promote appropriate growth. How would clinicians use to adjust and individual care for each infant? Likewise, how would clinicians use protein oxidation values in real-time clinical care?
Thank you for your comment. We add an explanation of the importance to have an adequate protein oxidation and modulate non protein energy/protein ratio in order to stimulate fat-free mass deposition.
I agree with your reported limitations. A huge consideration (which you included) is that there was no inclusion of infants with chronic conditions (e.g. bronchopulmonary dysplasia, etc.) who would have altered needs.
Thank you for your comment.

Round 2
Reviewer 1 Report
Sorry to say that in their revised version the authors did not address two fundamental concerns which I have raised in my 1st review.
First, since the REE on bw (or REE on FFM) regression has a non-zero intercept it is a fundamental flaw to present the data as simple ratios (i.e., as REE/kgbw or REE/kgFFM). There is need of a regression analysis to present the data as REE adjusted for either bw or FFM. This is a mandatory task
Second, the authors present pp data only, thus, they have measured REE + DIT. Since DIT starts immediately after food intake (i.e., it has an obligatory as well as a facultative component where the latter is due to central mechanisms thus it does not depend on the absorption of macronutrients) the authors rebuttal cannot be accepted. The paper has to be re-written accordingly.
Author Response
Sorry to say that in their revised version the authors did not address two fundamental concerns which I have raised in my 1st review.
First, since the REE on bw (or REE on FFM) regression has a non-zero intercept it is a fundamental flaw to present the data as simple ratios (i.e., as REE/kgbw or REE/kgFFM). There is need of a regression analysis to present the data as REE adjusted for either bw or FFM. This is a mandatory task
Thank you for you observation. In the first version we have not included regression analysis because our groups was homogeneous for basic characteristics (table 1-2). These data are confirmed after correction by GA and BW (beta 0.043, p=0.58) and without correction (beta 0.042, p=.0.62). Anyway, we added regression analysis in text, as you can see at line 225-227. We decided to report only data about regression analysis corrected for GA and BW.
Second, the authors present pp data only, thus, they have measured REE + DIT. Since DIT starts immediately after food intake (i.e., it has an obligatory as well as a facultative component where the latter is due to central mechanisms thus it does not depend on the absorption of macronutrients) the authors rebuttal cannot be accepted. The paper has to be re-written accordingly.
Thank you for you observation. In a previous paper published by our work-team, we have demonstrated that DIT not influences REE in VLBWI as REE/kg was similar in different time slots (from 20 min to 2 hours after meal): probably this is the consequence of the limit amount of milk assumed in a single meal by a VLBWI, that has a non-significant influences on REE. In addition, considering the frequency of meal in VLBWI (every 3 hours), DIT could be considered as a constant during the 24-hours. Moreover, in literature are reported that DIT start an hour after meal ingestion in adults (for example, see Tomassi G., Merendino N. (2006) Diet-Induced Thermogenesis. In: Mantovani G. et al. (eds) Cachexia and Wasting: A Modern Approach. Springer, Milano. https://doi.org/10.1007/978-88-470-0552-5_6). If we supposed an influences of nutrients and energy intake on REE immediately after meal as reported by Westerterp, we want to point out that our two groups are similar in term of protein, carbohydrates and lipid intake as energy amount, so we supposed that DIT was not statistically different into groups (see: Westerterp, K.R. Diet induced thermogenesis. Nutr Metab (Lond) 1, 5 (2004). https://doi.org/10.1186/1743-7075-1-5)

Reviewer 2 Report
After review of the changes, my few comments are as follows:- Line 115: Please change "kcal/kg/die" to "kcal/kg/day"
- Table 2, for "GA weeks" for the PFF group, I still think "1,44" likely needs to be changed to "1.44". Please check for accuracy.
- Line 316: I think this needs changed from "to adequate" to "to provide adequate".
- Per my previous comment, I would like to see greater detail in the discussion section for how clinicians can/should use values from the REE? This is resting energy expenditure (low at ~60 kcal/kg/day), but this is not adequate to promote appropriate growth. How would clinicians use to adjust and individual care for each infant? Likewise, how would clinicians use protein oxidation values in real-time clinical care? Assessing total calories/kg/day is not the same as assessing ratios of grams of protein vs. non-protein energy.